# SEED RL: Scalable and Efficient Deep-RL with Accelerated Central Inference

**Lasse Espeholt,**\* **Raphaël Marinier,**\* **Piotr Stanczyk,**\* **Ke Wang & Marcin Michalski**
Brain Team
Google Research
{lespeholt, raphaelm, stanczyk, kewa, michalski}@google.com

## ABSTRACT

We present a modern scalable reinforcement learning agent called SEED (Scalable, Efficient Deep-RL). By effectively utilizing modern accelerators, we show that it is not only possible to **train on millions of frames per second** but also to **lower the cost** of experiments compared to current methods. We achieve this with a simple architecture that features centralized inference and an optimized communication layer. SEED adopts two state of the art distributed algorithms, IMPALA/V-trace (policy gradients) and R2D2 (Q-learning), and is evaluated on Atari-57, DeepMind Lab and Google Research Football. We improve the state of the art on Football and are able to reach state of the art on Atari-57 three times faster in wall-time. For the scenarios we consider, a 40% to 80% cost reduction for running experiments is achieved. The implementation along with experiments is **open-sourced** so results can be reproduced and novel ideas tried out.

Github: `http://github.com/google-research/seed_rl`.

## 1 INTRODUCTION

The field of reinforcement learning (RL) has recently seen impressive results across a variety of tasks. This has in part been fueled by the introduction of deep learning in RL and the introduction of accelerators such as GPUs. In the very recent history, focus on massive scale has been key to solve a number of complicated games such as AlphaGo (Silver et al., 2016), Dota (OpenAI, 2018) and StarCraft 2 (Vinyals et al., 2017).

The sheer amount of environment data needed to solve tasks trivial to humans, makes distributed machine learning unavoidable for fast experiment turnaround time. RL is inherently comprised of heterogeneous tasks: running environments, model inference, model training, replay buffer, etc. and current state-of-the-art distributed algorithms do not efficiently use compute resources for the tasks. The amount of data and inefficient use of resources makes experiments unreasonably expensive. The two main challenges addressed in this paper are scaling of reinforcement learning and optimizing the use of modern accelerators, CPUs and other resources.

We introduce SEED (Scalable, Efficient, Deep-RL), a modern RL agent that scales well, is flexible and efficiently utilizes available resources. It is a distributed agent where model inference is done centrally combined with fast streaming RPCs to reduce the overhead of inference calls. We show that with simple methods, one can achieve state-of-the-art results faster on a number of tasks. For optimal performance, we use TPUs (cloud.google.com/tpu/) and TensorFlow 2 (Abadi et al., 2015) to simplify the implementation. The cost of running SEED is analyzed against IMPALA (Espeholt et al., 2018) which is a commonly used state-of-the-art distributed RL algorithm (Veeriah et al. (2019); Li et al. (2019); Deverett et al. (2019); Omidshafiei et al. (2019); Vezhnevets et al. (2019); Hansen et al. (2019); Schaarschmidt et al.; Tirumala et al. (2019), ...). We show cost reductions of up to 80% while being significantly faster. When scaling SEED to many accelerators, it can train on millions of frames per second. Finally, the implementation is open-sourced together with examples of running it at scale on Google Cloud (see Appendix A.4 for details) making it easy to reproduce results and try novel ideas.

---

\*Equal contribution

## 2    RELATED WORK

For value-based methods, an early attempt for scaling DQN was Nair et al. (2015) that used asynchronous SGD (Dean et al., 2012) together with a distributed setup consisting of actors, replay buffers, parameter servers and learners. Since then, it has been shown that asynchronous SGD leads to poor sample complexity while not being significantly faster (Chen et al., 2016; Espeholt et al., 2018). Along with advances for Q-learning such as prioritized replay (Schaul et al., 2015), dueling networks (Wang et al., 2016), and double-Q learning (van Hasselt, 2010; Van Hasselt et al., 2016) the state-of-the-art distributed Q-learning was improved with Ape-X (Horgan et al., 2018). Recently, R2D2 (Kapturowski et al., 2018) achieved impressive results across all the Arcade Learning Environment (ALE) (Bellemare et al., 2013) games by incorporating value-function rescaling (Pohlen et al., 2018) and LSTMs (Hochreiter & Schmidhuber, 1997) on top of the advancements of Ape-X.

There have also been many approaches for scaling policy gradients methods. A3C (Mnih et al., 2016) introduced asynchronous single-machine training using asynchronous SGD and relied exclusively on CPUs. GPUs were later introduced in GA3C (Mahmood, 2017) with improved speed but poor convergence results due to an inherently on-policy method being used in an off-policy setting. This was corrected by V-trace (Espeholt et al., 2018) in the IMPALA agent both for single-machine training and also scaled using a simple actor-learner architecture to more than a thousand machines. PPO (Schulman et al., 2017) serves a similar purpose to V-trace and was used in OpenAI Rapid (Petrov et al., 2018) with the actor-learner architecture extended with Redis (redis.io), an in-memory data store, and was scaled to 128,000 CPUs. For inexpensive environments like ALE, a single machine with multiple accelerators can achieve results quickly (Stooke & Abbeel, 2018). This approach was taken a step further by converting ALE to run on a GPU (Dalton et al., 2019).

A third class of algorithms is evolutionary algorithms. With simplicity and massive scale, they have achieved impressive results on a number of tasks (Salimans et al., 2017; Such et al., 2017).

Besides algorithms, there exist a number of useful libraries and frameworks for reinforcement learning. ELF (Tian et al., 2017) is a framework for efficiently interacting with environments, avoiding Python global-interpreter-lock contention. Dopamine (Castro et al., 2018) is a flexible research focused RL framework with a strong emphasis on reproducibility. It has state of the art agent implementations such as Rainbow (Hessel et al., 2017) but is single-threaded. TF-Agents (Guadarrama et al., 2018) and rlpyt (Stooke & Abbeel, 2019) both have a broader focus with implementations for several classes of algorithms but as of writing, they do not have distributed capability for large-scale RL. RLLib (Liang et al., 2017) provides a number of composable distributed components and a communication abstraction with a number of algorithm implementations such as IMPALA and Ape-X. Concurrent with this work, TorchBeast (Küttler et al., 2019) was released which is an implementation of single-machine IMPALA with remote environments.

SEED is closest related to IMPALA, but has a number of key differences that combine the benefits of single-machine training with a scalable architecture. Inference is moved to the learner but environments run remotely. This is combined with a fast communication layer to mitigate latency issues from the increased number of remote calls. The result is significantly faster training at reduced costs by as much as 80% for the scenarios we consider. Along with a policy gradients (V-trace) implementation we also provide an implementation of state of the art Q-learning (R2D2). In the work we use TPUs but in principle, any modern accelerator could be used in their place. TPUs are particularly well-suited given they high throughput for machine learning applications and the scalability. Up to 2048 cores are connected with a fast interconnect providing 100+ petaflops of compute.

## 3    ARCHITECTURE

Before introducing the architecture of SEED, we first analyze the generic actor-learner architecture used by IMPALA, which is also used in various forms in Ape-X, OpenAI Rapid and others. An overview of the architecture is shown in Figure 1a.

A large number of actors repeatedly read model parameters from the learner (or parameter servers). Each actor then proceeds the local model to sample actions and generate a full trajectory of observations, actions, policy logits/Q-values. Finally, this trajectory along with recurrent state is transferred

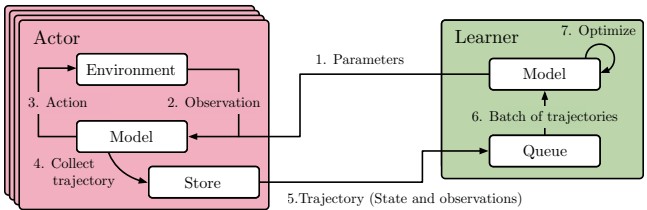

(a) IMPALA architecture (distributed version)

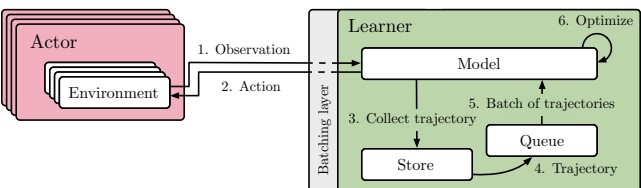

(b) SEED architecture, see detailed replay architecture in Figure 3.

Figure 1: Overview of architectures

to a shared queue or replay buffer. Asynchronously, the learner reads batches of trajectories from the queue/replay buffer and optimizes the model.

There are a number of reasons for why this architecture falls short:

1. **Using CPUs for neural network inference:** The actor machines are usually CPU-based (occasionally GPU-based for expensive environments). CPUs are known to be computationally inefficient for neural networks (Raina et al., 2009). When the computational needs of a model increase, the time spent on inference starts to outweigh the environment step computation. The solution is to increase the number of actors which increases the cost and affects convergence (Espeholt et al., 2018).

2. **Inefficient resource utilization:** Actors alternate between two tasks: environment steps and inference steps. The compute requirements for the two tasks are often not similar which leads to poor utilization or slow actors. E.g. some environments are inherently single-threading while neural networks are easily parallelizable.

3. **Bandwidth requirements:** Model parameters, recurrent state and observations are transferred between actors and learners. Relatively to model parameters, the size of the observation trajectory often only accounts for a few percents.[1] Furthermore, memory-based models send large states, increase bandwidth requirements.

While single-machine approaches such as GA3C (Mahmood, 2017) and single-machine IMPALA avoid using CPU for inference (1) and do not have network bandwidth requirements (3), they are restricted by resource usage (2) and the scale required for many types of environments.

The architecture used in SEED (Figure 1b) solves the problems mentioned above. Inference and trajectory accumulation is moved to the learner which makes it conceptually a single-machine setup with remote environments (besides handling failures). Moving the logic effectively makes the actors a small loop around the environments. For every single environment step, the observations are sent to the learner, which runs the inference and sends actions back to the actors. This introduces a new problem: 4. **Latency.**

To minimize latency, we created a simple framework that uses gRPC (grpc.io) - a high performance RPC library. Specifically, we employ streaming RPCs where the connection from actor to learner is kept open and metadata sent only once. Furthermore, the framework includes a batching module that efficiently batches multiple actor inference calls together. In cases where actors can fit on the same machine as learners, gRPC uses unix domain sockets and thus reduces latency, CPU and syscall overhead. Overall, the end-to-end latency, including network and inference, is faster for a number of the models we consider (see Appendix A.7).

---

[1]With 100,000 observations send per second (96 x 72 x 3 bytes each), a trajectory length of 20 and a 30MB model, the total bandwidth requirement is 148 GB/s. Transferring observations uses only 2 GB/s.

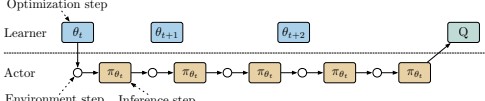

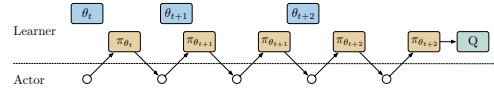

(a) Off-policy in IMPALA. For the entire trajectory the policy stays the same. By the time the trajectory is sent to the queue for optimization, the policy has changed twice.

(b) Off-policy in SEED. Optimizing a model has immediate effect on the policy. Thus, the trajectory consists of actions sampled from many different policies ($\pi_{\theta_t}$, $\pi_{\theta_{t+1}}$, ...).

Figure 2: Variants of "near on-policy" when evaluating a policy $\pi$ while asynchronously optimizing model parameters $\theta$.

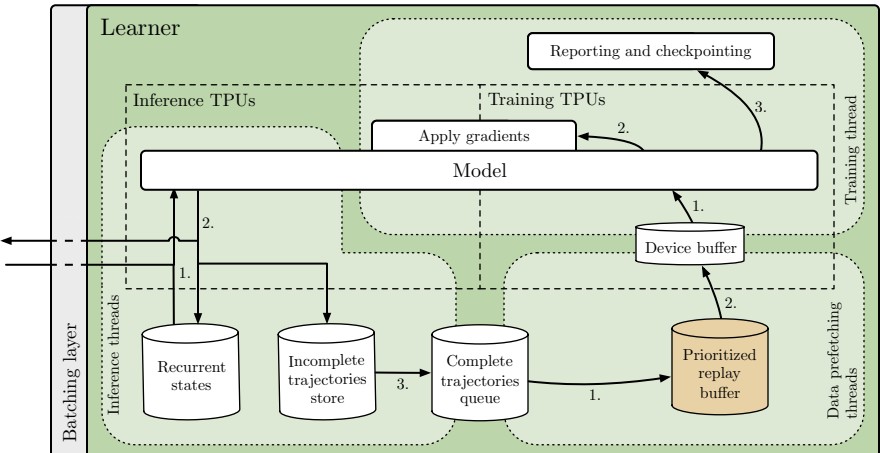

Figure 3: Detailed Learner architecture in SEED (with an optional replay buffer).

The IMPALA and SEED architectures differ in that for SEED, at any point in time, only one copy of the model exists whereas for distributed IMPALA each actor has its own copy. This changes the way the trajectories are off-policy. In IMPALA (Figure 2a), an actor uses the same policy $\pi_{\theta_t}$ for an entire trajectory. For SEED (Figure 2b), the policy during an unroll of a trajectory may change multiple times with later steps using more recent policies closer to the one used at optimization time.

A detailed view of the learner in the SEED architecture is shown on Figure 3. Three types of threads are running: 1. Inference 2. Data prefetching and 3. Training. Inference threads receive a batch of observations, rewards and episode termination flags. They load the recurrent states and send the data to the inference TPU core. The sampled actions and new recurrent states are received, and the actions are sent back to the actors while the latest recurrent states are stored. When a trajectory is fully unrolled it is added to a FIFO queue or replay buffer and later sampled by data prefetching threads. Finally, the trajectories are pushed to a device buffer for each of the TPU cores taking part in training. The training thread (the main Python thread) takes the prefetched trajectories, computes gradients using the training TPU cores and applies the gradients on the models of all TPU cores (inference and training) synchronously. The ratio of inference and training cores can be adjusted for maximum throughput and utilization. The architecture scales to a TPU pod (2048 cores) by round-robin assigning actors to TPU host machines, and having separate inference threads for each TPU host. When actors wait for a response from the learner, they are idle so in order to fully utilize the machines, we run multiple environments on a single actor.

To summarize, we solve the issues listed previously by:

1. Moving inference to the learner and thus eliminating any neural network related computations from the actors. Increasing the model size in this architecture will not increase the need for more actors (in fact the opposite is true).

2. Batching inference on the learner and having multiple environments on the actor. This fully utilize both the accelerators on the learner and CPUs on the actors. The number of

TPU cores for inference and training is finely tuned to match the inference and training workloads. All factors help reducing the cost of experiments.

3. Everything involving the model stays on the learner and only observations and actions are sent between the actors and the learner. This reduces bandwidth requirements by as much as 99%.

4. Using streaming gRPC that has minimal latency and minimal overhead and integrating batching into the server module.

We provide the following two algorithms implemented in the SEED framework: V-trace and Q-learning.

## 3.1 V-TRACE

One of the algorithms we adapt into the framework is V-trace (Espeholt et al., 2018). We do not include any of the additions that have been proposed on top of IMPALA such as van den Oord et al. (2018); Gregor et al. (2019). The additions can also be applied to SEED and since they are more computational expensive, they would benefit from the SEED architecture.

## 3.2 Q-LEARNING

We show the versatility of SEED's architecture by fully implementing R2D2 (Kapturowski et al., 2018), a state of the art distributed value-based agent. R2D2 itself builds on a long list of improvements over DQN (Mnih et al., 2015): double Q-learning (van Hasselt, 2010; Van Hasselt et al., 2016), multi-step bootstrap targets (Sutton, 1988; Sutton & Barto, 1998; Mnih et al., 2016), dueling network architecture (Wang et al., 2016), prioritized distributed replay buffer (Schaul et al., 2015; Horgan et al., 2018), value-function rescaling (Pohlen et al., 2018), LSTM's (Hochreiter & Schmidhuber, 1997) and burn-in (Kapturowski et al., 2018).

Instead of a distributed replay buffer, we show that it is possible to keep the replay buffer on the learner with a straightforward flexible implementation. This reduces complexity by removing one type of job in the setup. It has the drawback of being limited by the memory of the learner but it was not a problem in our experiments by a large margin: a replay buffer of $10^5$ trajectories of length 120 of $84 \times 84$ uncompressed grayscale observations (following R2D2's hyperparameters) takes 85GBs of RAM, while Google Cloud machines can offer hundreds of GBs. However, nothing prevents the use of a distributed replay buffer together with SEED's central inference, in cases where a much larger replay buffer is needed.

## 4 EXPERIMENTS

We evaluate SEED on a number of environments: DeepMind Lab (Beattie et al., 2016), Google Research Football (Kurach et al., 2019) and Arcade Learning Environment (Bellemare et al., 2013).

## 4.1 DEEPMIND LAB AND V-TRACE

DeepMind Lab is a 3D environment based on the Quake 3 engine. It features mazes, laser tag and memory tasks. We evaluate on four commonly used tasks. The action set used is from Espeholt et al. (2018) although for some tasks, higher return can be achieved with bigger action sets such as the one introduced in Hessel et al. (2018). For all experiments, we used an action repeat of 4 and the number of frames in plots is listed as environment frames (equivalent to 4 times the number of steps). The same set of 24 hyperparameter sets and the same model (ResNet from IMPALA) was used for both agents. More details can be found in Appendix A.1.2.

### 4.1.1 STABILITY

The first experiment evaluates the effect of the change in off-policy behavior described in Figure 2. Exactly the same hyperparameters are used for both IMPALA and SEED, including the number of environments used. As is shown in Figure 4, the stability across hyperparameters of SEED is slightly better than IMPALA, while achieving slightly higher final returns.

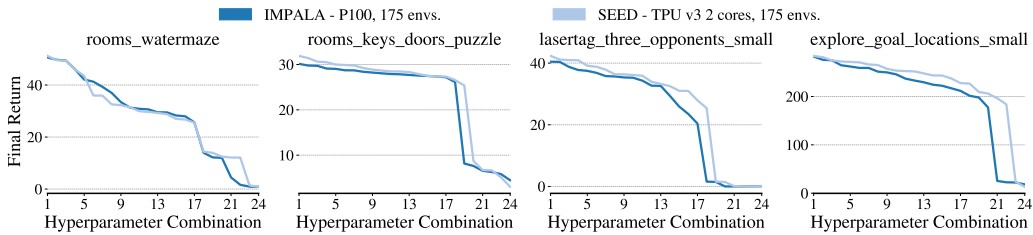

Figure 4: Comparison of IMPALA and SEED under the exact same conditions (175 actors, same hyperparameters, etc.) The plots show hyperparameter combinations sorted by the final performance across different hyperparameter combinations.

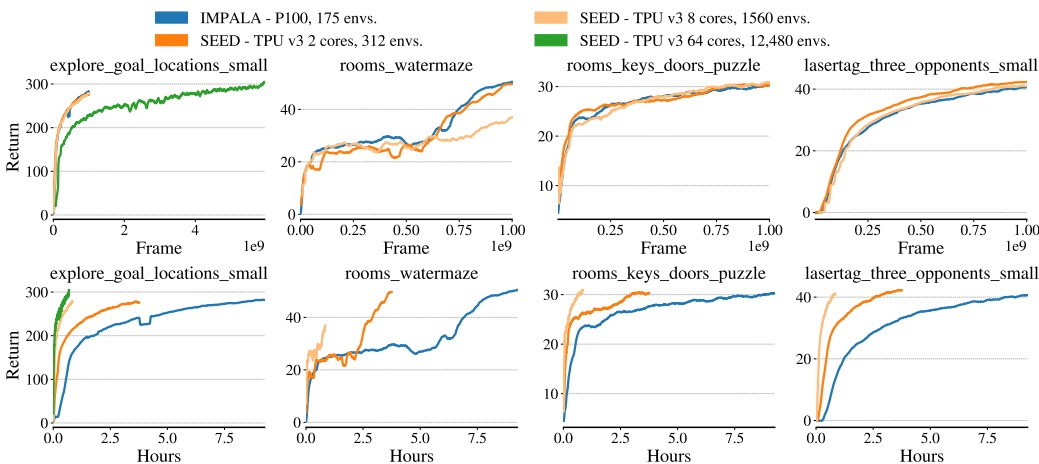

Figure 5: Training on 4 DeepMind Lab tasks. Each curve is the best of the 24 runs based on final return following the evaluation procedure in Espeholt et al. (2018). Sample complexity is maintained up to 8 TPU v3 cores, which leads to 11x faster training than the IMPALA baseline. **Top Row:** X-axis is per frame (number of frames = 4x number of steps). **Bottom Row:** X-axis is hours.

### 4.1.2 SPEED

For evaluating performance, we compare IMPALA using an Nvidia P100 with SEED with multiple accelerator setups. They are evaluated on the same set of hyperparameters. We find that SEED is 2.5x faster than IMPALA using 2 TPU v3 cores (see Table 1), while using only 77% more environments and 41% less CPU (see section 4.4.1). Scaling from 2 to 8 cores results in an additional 4.4x speedup with sample complexity maintained (Figure 5). The speed-up is greater than 4x due to using 6 cores for training and 2 for inference instead of 1 core for each, resulting in better utilization. A **5.3x** speed-up instead of 4.4x can be obtained by increasing the batch size linearly with the number of training cores, but contrary to related research (You et al., 2017b; Goyal et al., 2017) we found that increased batch size hurts sample complexity even with methods like warm-up and actor de-correlation (Stooke & Abbeel, 2018). We hypothesize that this is due to the limited actor and environment diversity in DeepMind Lab tasks. In McCandlish et al. (2018) they found that Pong scales poorly with batch size but that Dota can be trained effectively with a batch size five orders of magnitude larger. Note, for most models, the effective batch size is $\text{batch size} \cdot \text{trajectory length}$. In Figure 5, we include a run from a limited sweep on "explore_goal_locations_small" using 64 cores with an almost linear speed-up. Wall-time performance is improved but sample complexity is heavily penalized.

When using an Nvidia P100, SEED is 1.58x slower than IMPALA. A slowdown is expected because SEED performs inference on the accelerator. SEED does however use significantly fewer CPUs and is lower cost (see Appendix A.6). The TPU version of SEED has been optimized but it is likely that improvements can be found for SEED with P100.

| Architecture | Accelerators | Environments | Actor CPUs | Batch Size | FPS | Ratio |
|---|---|---:|---:|---:|---:|---|
| **DeepMind Lab** | | | | | | |
| IMPALA | Nvidia P100 | 176 | 176 | 32 | 30K | — |
| SEED | Nvidia P100 | 176 | 44 | 32 | 19K | **0.63x** |
| SEED | TPU v3, 2 cores | 312 | 104 | 32 | 74K | **2.5x** |
| SEED | TPU v3, 8 cores | 1560 | 520 | $48^1$ | 330K | **11.0x** |
| SEED | TPU v3, 64 cores | 12,480 | 4,160 | $384^1$ | 2.4M | **80.0x** |
| **Google Research Football** | | | | | | |
| IMPALA, Default | 2 x Nvidia P100 | 400 | 400 | 128 | 11K | — |
| SEED, Default | TPU v3, 2 cores | 624 | 416 | 128 | 18K | **1.6x** |
| SEED, Default | TPU v3, 8 cores | 2,496 | 1,664 | $160^3$ | 71K | **6.5x** |
| SEED, Medium | TPU v3, 8 cores | 1,550 | 1,032 | $160^3$ | 44K | — |
| SEED, Large | TPU v3, 8 cores | 1,260 | 840 | $160^3$ | 29K | — |
| SEED, Large | TPU v3, 32 cores | 5,040 | 3,360 | $640^3$ | 114K | **3.9x** |
| **Arcade Learning Environment** | | | | | | |
| R2D2 | Nvidia V100 | 256 | N/A | 64 | $85K^2$ | — |
| SEED | Nvidia V100 | 256 | 55 | 64 | 67K | **0.79x** |
| SEED | TPU v3, 8 cores | 610 | 213 | 64 | 260K | **3.1x** |
| SEED | TPU v3, 8 cores | 1200 | 419 | 256 | $440K^4$ | **5.2x** |

[1] 6/8 cores used for training. [2] Each of the 256 R2D2 actors run at 335 FPS (information from the R2D2 authors). [3] 5/8 cores used for training. [4] No frame stacking.

Table 1: Performance of SEED, IMPALA and R2D2.

## 4.2 GOOGLE RESEARCH FOOTBALL AND V-TRACE

Google Research Football is an environment similar to FIFA video games (ea.com). We evaluate SEED on the "Hard" task introduced in Kurach et al. (2019) where the baseline IMPALA agent achieved a positive average score after 500M frames using the engineered "checkpoint" reward function but a negative average score using only the score as a reward signal. In all experiments we use the model from Kurach et al. (2019) and sweep over 40 hyperparameter sets with 3 seeds each. See all hyperparameters in Appendix A.2.1.

### 4.2.1 SPEED

Compared to the baseline IMPALA using 2 Nvidia P100's (and CPUs for inference) in Kurach et al. (2019) we find that using 2 TPU v3 cores in SEED improves the speed by 1.6x (see Table 1). Additionally, using 8 cores adds another 4.1x speed-up. A speed-up of **4.5x** is achievable if the batch size is increased linearly with the number of training cores (5). However, we found that increasing the batch size, like with DeepMind Lab, hurts sample complexity.

### 4.2.2 INCREASED MAP SIZE

More compute power allows us to increase the size of the Super Mini Map (SMM) input state. By default its size is 96 x 72 (x 4) and it represents players, opponents, ball and the active player as 2d bit maps. We evaluated three sizes: (1) Default 96 x 72, (2) Medium 120 x 90 and (3) Large 144 x 108. As shown in Table 1, switching from Default to Large SMM results in 60% speed reduction. However, increasing the map size improves the final score (Table 2). It may suggest that the bit map representation is not granular enough for the task. For both reward functions, we improve upon the results of Kurach et al. (2019). Finally, training on 4B frames improves the results by a significant margin (an average score of 0.46 vs. 4.76 in case of the scoring reward function).

## 4.3 ARCADE LEARNING ENVIRONMENT AND Q-LEARNING

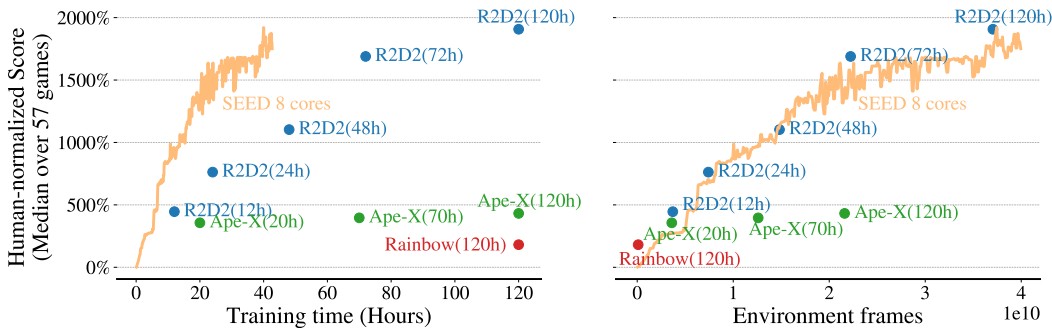

Figure 6: Median human-normalized score on Atari-57 for SEED and related agents. SEED was run with 1 seed for each game. All agents use up to 30 random no-ops for evaluation. **Left:** X-axis is hours **Right:** X-axis is environment frames (a frame is 1/4th of an environment step due to action repeat). SEED reaches state of the art performance **3.1x** faster (wall-time) than R2D2.

We evaluate our implementation of R2D2 in SEED architecture on 57 Atari 2600 games from the ALE benchmark. This benchmark has been the testbed for most recent deep reinforcement learning agents because of the diversity of visuals and game mechanics.

We follow the same evaluation procedure as R2D2. In particular, we use the full action set, no loss-of-life-as-episode-end heuristic and start episodes with up to 30 random no-ops. We use 8 TPU v3 cores and 610 actors to maximize TPU utilization. This achieves 260K environment FPS and performs 9.5 network updates per second. Other hyperparameters are taken from R2D2, and are fully reproduced in appendix A.3.1.

Figure 6 shows the median human-normalized scores for SEED, R2D2, Ape-X and Rainbow. As expected, SEED has similar sample efficiency as R2D2, but it is **3.1x** faster (see Table 1). This allows us to reach a median human-normalized score of 1880% in just 1.8 days of training instead of 5, establishing a new wall-time state of the art on Atari-57.

With the number of actors increased to 1200, a batch size increased to 256 and without frame-stacking, we can achieve 440K environment FPS and learn using 16

| Architecture | Accelerators | SMM | Median | Max |
|---|---|---|---|---|
| **Scoring reward** | | | | |
| IMPALA | 2 x Nvidia P100 | Default | -0.74 | 0.06 |
| SEED | TPU v3, 2 cores | Default | -0.72 | -0.12 |
| SEED | TPU v3, 8 cores | Default | -0.83 | -0.02 |
| SEED | TPU v3, 8 cores | Medium | -0.74 | 0.12 |
| SEED | TPU v3, 8 cores | Large | **-0.69** | **0.46** |
| SEED | TPU v3, 32 cores | Large | n/a | **4.76**[1] |
| **Checkpoint reward** | | | | |
| IMPALA | 2 x Nvidia P100 | Default | **-0.27** | 1.63 |
| SEED | TPU v3, 2 cores | Default | -0.44 | 1.64 |
| SEED | TPU v3, 8 cores | Default | -0.68 | 1.55 |
| SEED | TPU v3, 8 cores | Medium | -0.52 | 1.76 |
| SEED | TPU v3, 8 cores | Large | -0.38 | **1.86** |
| SEED | TPU v3, 32 cores | Large | n/a | **7.66**[1] |

[1] 32 core experiments trained on 4B frames with a limited sweep.

Table 2: Google Research Football "Hard" using two kinds of reward functions. For each reward function, 40 hyperparameter sets ran with 3 seeds each which were averaged after 500M frames of training. The table shows the median and maximum of the 40 averaged values. This is a similar setup to Kurach et al. (2019) although we ran 40 hyperparameter sets vs. 100 but did not rerun our best models using 5 seeds.

batches per second. However, this significantly degrades sample efficiency and limits the final median human-normalized score to approximately 1000%.

## 4.4 COST COMPARISONS

With growing complexity of environments as well as size of neural networks used in reinforcement learning, the need of running big experiments increases, making cost reductions important. In this

section we analyze how increasing complexity of the network impacts training cost for SEED and IMPALA. In our experiments we use the pricing model of Google AI Platform, ML Engine.[2]

Our main focus is on obtaining lowest possible cost per step, while maintaining training speed. Distributed experiments from Espeholt et al. (2018) (IMPALA) used between 150 and 500 CPUs, which translates into $7.125 - $23.75 of actors' cost per hour. The cost of single-GPU learner is $1.46 per hour. Due to the relatively high expense of the actors, our main focus is to reduce number of actors and to obtain high CPU utilization.

| Resource | Cost per hour |
|---|---|
| CPU core | $0.0475 |
| Nvidia Tesla P100 | $1.46 |
| TPU v3 core | $1.00 |

Table 3: Cost of cloud resources as of Sep. 2019.

### 4.4.1 DEEPMIND LAB

Our DeepMind Lab experiment is based on the ResNet model from IMPALA. We evaluate increasing the number of filters in the convolutional layers: (1) Default 1x (2) Medium 2x and (3) Large 4x. Experiments are performed on the "explore_goal_locations_small" task. IMPALA uses a single Nvidia Tesla P100 GPU for training while inference is done on CPU by the actors. SEED uses one TPU v3 core for training and one for inference.

For IMPALA, actor CPU utilization is close to 100% but in case of SEED, only the environment runs on an actor making CPU idle while waiting for inference step. To improve utilization, a single SEED actor runs multiple environments (between 12 and 16) on a 4-CPU machine.

| Model | Actors | CPUs | Envs. | Speed | Cost/1B | Cost ratio |
|---|---|---|---|---|---|---|
| **IMPALA** | | | | | | |
| Default | 176 | 176 | 176 | 30k | $90 | — |
| Medium | 130 | 130 | 130 | 16.5k | $128 | — |
| Large | 100 | 100 | 100 | 7.3k[1] | $236 | — |
| **SEED** | | | | | | |
| Default | 26 | 104 | 312 | 74k | $25 | **3.60** |
| Medium | 12 | 48 | 156 | 34k | $35 | **3.66** |
| Large | 6 | 24 | 84 | 16k | $54 | **4.37** |

[1] The batch size was lowered from 32 to 16 due to limited memory on Nvidia P100.

Table 4: Training cost on DeepMind Lab for 1 billion frames.

As Table 4 shows, SEED turns out to be not only faster, but also cheaper to run. The cost ratio between SEED and IMPALA is around 4. Due to high cost of inference on a CPU, IMPALA's cost increases with increasing complexity of the network. In the case of SEED, increased network size has smaller impact on overall costs, as inference accounts for about 30% of the costs (see Table A.5).

### 4.4.2 GOOGLE RESEARCH FOOTBALL

We evaluate cost of running experiments with Google Research Football with different sizes of the Super Mini Map representation (the size has virtually no impact on environment's speed.) For IMPALA, two Nvidia P100 GPUs were used for training and SEED used one TPU v3 core for training and one for inference.

For IMPALA, we use one core per actor while SEED's actors run multiple environments per actor for better CPU utilization (8 cores, 12 environments).

For the default size of the SMM, per experiment training cost differs by only 68%. As the Google Research Football environment is more expensive than DeepMind Lab, training and inference costs

---

[2]TPU cores are sold in multiples of 8, by running many experiments at once we use as many cores per experiment as needed. See cloud.google.com/ml-engine/docs/pricing.

| Model | Actors | CPUs | Envs. | Speed | Cost/1B | Cost ratio |
|---|---|---|---|---|---|---|
| **IMPALA** | | | | | | |
| Default | 400 | 400 | 400 | 11k | $553 | — |
| Medium | 300 | 300 | 300 | 7k | $681 | — |
| Large | 300 | 300 | 300 | 5.3k | $899 | — |
| **SEED** | | | | | | |
| Default | 52 | 416 | 624 | 17.5k | $345 | **1.68** |
| Medium | 31 | 248 | 310 | 10.5k | $365 | **1.87** |
| Large | 21 | 168 | 252 | 7.5k | $369 | **2.70** |

Table 5: Training cost on Google Research Football for 1 billion frames.

have relatively smaller impact on the overall experiment cost. The difference increases when the size of the SMM increases and SEED needing relatively fewer actors.

### 4.4.3 ARCADE LEARNING ENVIRONMENT

Due to lack of baseline implementation for R2D2, we do not include cost comparisons for this environment. However, comparison of relative costs between ALE, DeepMind Lab and Football suggests that savings should be even more significant. ALE is the fastest among the three environments making inference proportionally the most expensive. Appendix A.5 presents training cost split between actors and the learner for different setups.

## 5 CONCLUSION

We introduced and analyzed a new reinforcement learning agent architecture that is faster and less costly per environment frame than previous distributed architectures by better utilizing modern accelerators. It achieved a 11x wall-time speedup on DeepMind Lab compared to a strong IMPALA baseline while keeping the same sample efficiency, improved on state of the art scores on Google Research Football, and achieved state of the art scores on Atari-57 3.1x faster (wall-time) than previous research.

The agent is open-sourced and packaged to easily run on Google Cloud. We hope that this will accelerate reinforcement learning research by allowing the community to replicate state-of-the-art results and build on top of them.

As a demonstration of the potential of this new agent architecture, we were able to scale it to millions of frames per second in some realistic scenarios (80x speedup compared to previous research). However, this requires increasing the number of environments and using larger batch sizes which hurts sample efficiency in the environments tested. Preserving sample efficiency with larger batch-sizes has been studied to some extent in RL (Stooke & Abbeel, 2018; McCandlish et al., 2018) and in the context of supervised learning (You et al., 2017b;a; 2019; Goyal et al., 2017). We believe it is still an open and increasingly important area of research in order to scale up reinforcement learning.

### ACKNOWLEDGMENTS

We would like to thank Steven Kapturowski, Georg Ostrovski, Tim Salimans, Aidan Clark, Manuel Kroiss, Matthieu Geist, Leonard Hussenot, Alexandre Passos, Marvin Ritter, Neil Zeghidour, Marc G. Bellemare and Sylvain Gelly for comments and insightful discussions and Marcin Andrychowicz, Dan Abolafia, Damien Vincent, Dehao Chen, Eugene Brevdo and Ruoxin Sang for their code contributions.

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

## A  APPENDIX

### A.1  DEEPMIND LAB

#### A.1.1  LEVEL CACHE

We enable DeepMind Lab's option for using a level cache for both SEED and IMPALA which greatly reduces CPU usage and results in stable actor CPU usage at close to 100% for a single core.

#### A.1.2  HYPERPARAMETERS

| Parameter | Range |
|---|---:|
| Action Repetitions | 4 |
| Discount Factor ($\gamma$) | $\{.99, .993, .997, .999\}$ |
| Entropy Coefficient | Log-uniform $(1e{-}5, 1e{-}3)$ |
| Learning Rate | Log-uniform $(1e{-}4, 1e{-}3)$ |
| Optimizer | Adam |
| Adam Epsilon | $\{1e{-}1, 1e{-}3, 1e{-}5, 1e{-}7\}$ |
| Unroll Length/$n$-step | 32 |
| Value Function Coefficient | .5 |
| V-trace $\lambda$ | $\{.9, .95, .99, 1.\}$ |

Table 6: Hyperparameter ranges used in the stability experiments.

## A.2 GOOGLE RESEARCH FOOTBALL

### A.2.1 HYPERPARAMETERS

| Parameter | Range |
|---|---|
| Action Repetitions | 1 |
| Discount Factor ($\gamma$) | $\{.99, .993, .997, .999\}$ |
| Entropy Coefficient | Log-uniform ($1e{-}7, 1e{-}3$) |
| Learning Rate | Log-uniform ($1e{-}5, 1e{-}3$) |
| Optimizer | Adam |
| Unroll Length/$n$-step | 32 |
| Value Function Coefficient | .5 |
| V-trace $\lambda$ | $\{.9, .95, .99, 1.\}$ |

Table 7: Hyperparameter ranges used for experiments with scoring and checkpoint rewards.

## A.3 ALE

### A.3.1 HYPERPARAMETERS

We use the same hyperparameters as R2D2 (Kapturowski et al., 2018), except that we use more actors in order to best utilize 8 TPU v3 cores. For completeness, agent hyperparameters are in table 8 and environment processing parameters in table 9. We use the same neural network architecture as R2D2, namely 3 convolutional layers with filter sizes $[32, 64, 64]$, kernel sizes $[8 \times 8, 4 \times 4, 3 \times 3]$ and strides $[4, 2, 1]$, ReLU activations and "valid" padding. They feed into a linear layer with 512 units, feeding into an LSTM layer with 512 hidden units (that also uses the one-hot encoded previous action and the previous environment reward as input), feeding into dueling heads with 512 hidden units. We use Glorot uniform  (Glorot & Bengio, 2010) initialization.

| Parameter | Value |
|---|---|
| Number of actors | 610 |
| Replay ratio | 0.75 |
| Sequence length | 120 incl. prefix of 40 burn-in transitions |
| Replay buffer size | $10^5$ part-overlapping sequences |
| Minimum replay buffer size | 5000 part-overlapping sequences |
| Priority exponent | 0.9 |
| Importance sampling exponent | 0.6 |
| Discount $\gamma$ | 0.997 |
| Training batch size | 64 |
| Inference batch size | 64 |
| Optimizer | Adam (lr $= 10^{-4}$, $\epsilon = 10^{-3}$)  (Kingma & Ba, 2014) |
| Target network update interval | 2500 updates |
| Value function rescaling | $x \mapsto \text{sign}(x)(\sqrt{|x| + 1} - 1) + \epsilon x, \epsilon = 10^{-3}$ |
| Gradient norm clipping | 80 |
| $n$-steps | 5 |
| Epsilon-greedy | **Training:** $i$-th actor $\in [0, N)$ uses $\epsilon_i = 0.4^{1 + \frac{7i}{N-1}}$ |
|  | **Evaluation:** $\epsilon = 10^{-3}$ |
| Sequence priority | $p = \eta \max_i \delta_i + (1 - \eta)\bar{\delta}$ where $\eta = 0.9$, |
|  | $\delta_i$ are per-step absolute TD errors. |

Table 8: SEED agent hyperparameters for Atari-57.

| Parameter | Value |
|---|---|
| Observation size | $84 \times 84$ |
| Resizing method | Bilinear |
| Random no-ops | uniform in $[1, 30]$. Applied before action repetition. |
| Frame stacking | 4 |
| Action repetition | 4 |
| Frame pooling | 2 |
| Color mode | grayscale |
| Terminal on loss of life | False |
| Max frames per episode | 108K (30 minutes) |
| Reward clipping | No |
| Action set | Full (18 actions) |
| Sticky actions | No |

Table 9: Atari-57 environment processing parameters.

A.3.2 FULL RESULTS ON ATARI-57

| Game | Human | R2D2 | SEED 8 TPU v3 cores |
|---|---|---|---|
| Alien | 7127.7 | 229496.9 | **262197.4** |
| Amidar | 1719.5 | **29321.4** | 28976.4 |
| Assault | 742.0 | **108197.0** | 102954.7 |
| Asterix | 8503.3 | **999153.3** | 983821.0 |
| Asteroids | 47388.7 | **357867.7** | 296783.0 |
| Atlantis | 29028.1 | **1620764.0** | 1612438.0 |
| BankHeist | 753.1 | 24235.9 | **47080.6** |
| BattleZone | 37187.5 | 751880.0 | **777200.0** |
| BeamRider | 16926.5 | **188257.4** | 173505.3 |
| Berzerk | 2630.4 | 53318.7 | **57530.4** |
| Bowling | 160.7 | **219.5** | 204.2 |
| Boxing | 12.1 | 98.5 | **100.0** |
| Breakout | 30.5 | 837.7 | **854.1** |
| Centipede | 12017.0 | **599140.3** | 574373.1 |
| ChopperCommand | 7387.8 | 986652.0 | **994991.0** |
| CrazyClimber | 35829.4 | **366690.7** | 337756.0 |
| Defender | 18688.9 | **665792.0** | 555427.2 |
| DemonAttack | 1971.0 | 140002.3 | **143748.6** |
| DoubleDunk | -16.4 | 23.7 | **24.0** |
| Enduro | 860.5 | **2372.7** | 2369.3 |
| FishingDerby | -38.7 | **85.8** | 75.0 |
| Freeway | 29.6 | 32.5 | **33.0** |
| Frostbite | 4334.7 | **315456.4** | 101726.8 |
| Gopher | 2412.5 | **124776.3** | 117650.4 |
| Gravitar | 3351.4 | **15680.7** | 7813.8 |
| Hero | 30826.4 | **39537.1** | 37223.1 |
| IceHockey | 0.9 | **79.3** | 79.2 |
| Jamesbond | 302.8 | 25354.0 | **25987.0** |
| Kangaroo | 3035.0 | **14130.7** | 13862.0 |
| Krull | 2665.5 | **218448.1** | 113224.8 |
| KungFuMaster | 22736.3 | 233413.3 | **239713.0** |
| MontezumaRevenge | **4753.3** | 2061.3 | 900.0 |
| MsPacman | 6951.6 | 42281.7 | **43115.4** |
| NameThisGame | 8049.0 | 58182.7 | **68836.2** |
| Phoenix | 7242.6 | 864020.0 | **915929.6** |
| Pitfall | **6463.7** | 0.0 | -0.1 |
| Pong | 14.6 | **21.0** | **21.0** |
| PrivateEye | **69571.3** | 5322.7 | 198.0 |
| Qbert | 13455.0 | 408850.0 | **546857.5** |
| Riverraid | 17118.0 | **45632.1** | 36906.4 |
| RoadRunner | 7845.0 | 599246.7 | **601220.0** |
| Robotank | 11.9 | 100.4 | **104.8** |
| Seaquest | 42054.7 | **999996.7** | 999990.2 |
| Skiing | **-4336.9** | -30021.7 | -29973.6 |
| Solaris | **12326.7** | 3787.2 | 861.6 |
| SpaceInvaders | 1668.7 | 43223.4 | **62957.8** |
| StarGunner | 10250.0 | **717344.0** | 448480.0 |
| Surround | 6.5 | **9.9** | 9.8 |
| Tennis | -8.3 | -0.1 | **23.9** |
| TimePilot | 5229.2 | **445377.3** | 444527.0 |
| Tutankham | 167.6 | **395.3** | 376.5 |
| UpNDown | 11693.2 | **589226.9** | 549355.4 |
| Venture | 1187.5 | 1970.7 | **2005.5** |
| VideoPinball | 17667.9 | **999383.2** | 979432.1 |
| WizardOfWor | 4756.5 | **144362.7** | 136352.5 |
| YarsRevenge | 54576.9 | **995048.4** | 973319.0 |
| Zaxxon | 9173.3 | **224910.7** | 168816.5 |

Table 10: Final performance of SEED 8 TPU v3 cores, 610 actors (1 seed) compared to R2D2 (averaged over 3 seeds) and Human, using up to 30 random no-op steps at the beginning of each episode. SEED was evaluated by averaging returns over 200 episodes for each game after training on 40e9 environment frames.

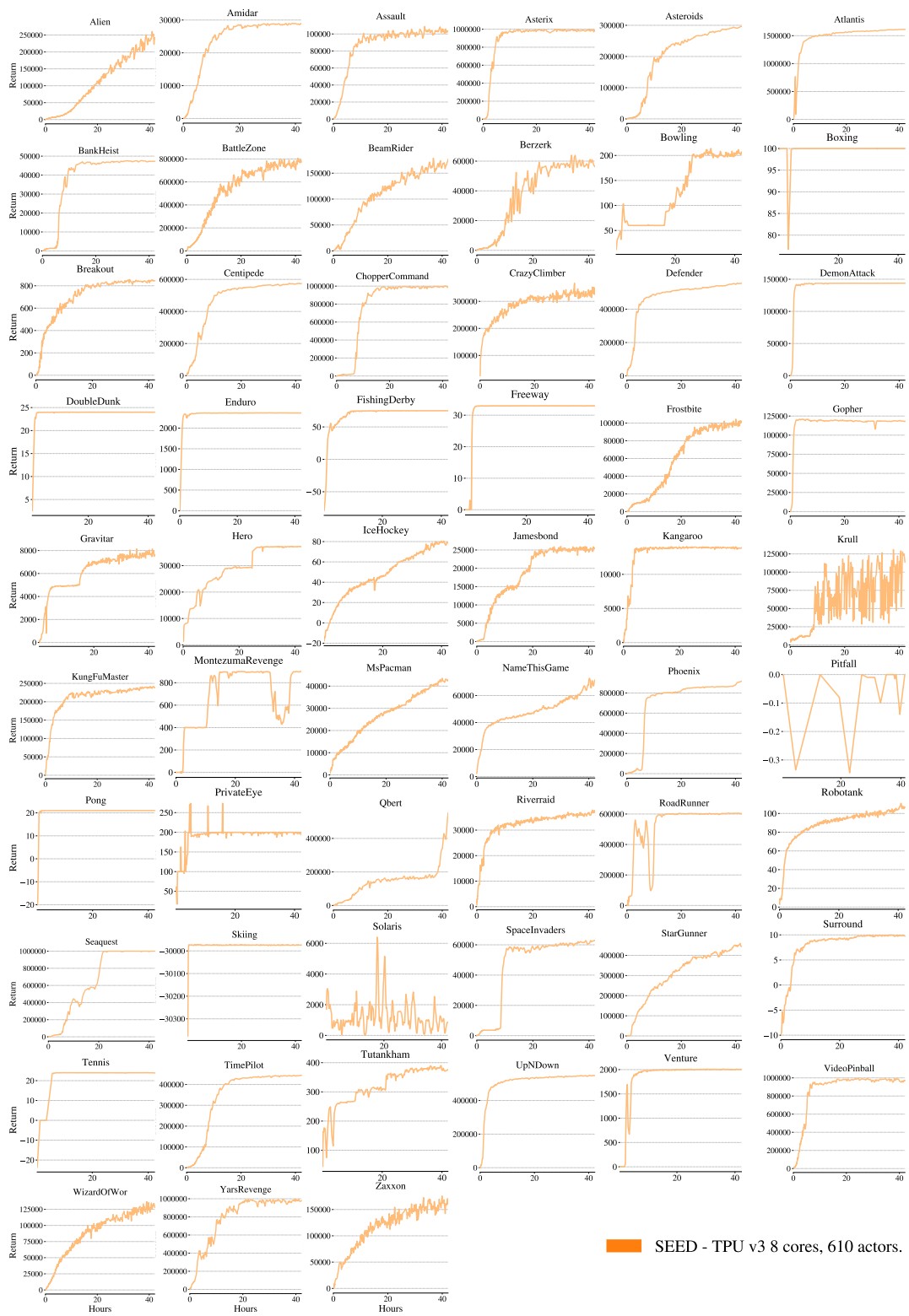

Figure 7: Learning curves on 57 Atari 2600 games for SEED (8 TPUv3 cores, 610 actors, evaluated with 1 seed). Each point of each curve averages returns over 200 episodes. No curve smoothing was performed. Curves end at approximately 43 hours of training, corresponding to $40e9$ environment frames.

## A.4    SEED LOCALLY AND ON CLOUD

SEED is open-sourced together with an example of running it both on a local machine and with scale using AI Platform, part of Google Cloud. We provide a public Docker image with low-level components implemented in C++ already pre-compiled to minimize the time needed to start SEED experiments.

The main pre-requisite to running on Cloud is setting up a Cloud Project. The provided startup script uploads the image and runs training for you. For more details please see `github.com/google-research/seed_rl`.

## A.5    EXPERIMENTS COST SPLIT

| Model | Algorithm | Actors cost | Learner cost | Total cost |
|-------|-----------|-------------|--------------|------------|
| **Arcade Learning Environment** | | | | |
| Default | SEED | $10.8 | $8.5 | $19.3 |
| **DeepMind Lab** | | | | |
| Default | IMPALA | $77.0 | $13.4 | $90 |
| Medium | IMPALA | $103.6 | $24.4 | $128 |
| Large | IMPALA | $180.5 | $55.5 | $236 |
| Default | SEED | $20.1 | $8.2 | $28 |
| Medium | SEED | $18.6 | $16.4 | $35 |
| Large | SEED | $19.6 | $35 | $54 |
| **Google Research Football** | | | | |
| Default | IMPALA | $479 | $74 | $553 |
| Medium | IMPALA | $565.2 | $115.8 | $681 |
| Large | IMPALA | $746.1 | $153 | $899 |
| Default | SEED | $313 | $32 | $345 |
| Medium | SEED | $312 | $53 | $365 |
| Large | SEED | $295 | $74 | $369 |

Table 11: Cost of performing 1 billion frames for both IMPALA and SEED split by component.

## A.6    COST COMPARISON ON DEEPMIND LAB USING NVIDIA P100 GPUS

In section 4.4.1, we compared the cost of running SEED using two TPU v3 cores and IMPALA on a single Nvidia P100 GPU. In table 12, we also compare the cost when both agents run on a single Nvidia P100 GPU on DeepMind Lab. Even though this is a sub-optimal setting for SEED because it performs inference on the accelerator and therefore benefits disproportionately from more efficient accelerators such as TPUs, SEED compares favorably, being **1.76x** cheaper than IMPALA per frame.

| Architecture | Actors | CPUs | Envs. | Speed | Cost/1B | Cost ratio |
|--------------|--------|------|-------|-------|---------|------------|
| IMPALA | 176 | 176 | 176 | 30k | $90 | — |
| SEED | 15 | 44 | 176 | 19k | $51 | **1.76** |

Table 12: Cost of performing 1 billion frames for both IMPALA and SEED when running on a single Nvidia P100 GPU on DeepMind Lab.

## A.7 INFERENCE LATENCY

| Model | IMPALA | SEED |
|-------|--------|------|
| **DeepMind Lab** | | |
| Default | 17.97ms | 10.98ms |
| Medium | 25.86ms | 12.70ms |
| Large | 48.79ms | 14.99ms |
| **Google Research Football** | | |
| Default | 12.59ms | 6.50ms |
| Medium | 19.24ms | 5.90ms |
| Large | 34.20ms | 11.19ms |
| **Arcade Learning Environment** | | |
| Default | N/A | 7.2ms |

Table 13: End-to-end inference latency of IMPALA and SEED for different environments and models.

