# OpenReview forum: "SEED RL: Scalable and Efficient Deep-RL with Accelerated Central Inference"
_ICLR.cc/2020/Conference — Accept (Talk)_

### Official Review · AnonReviewer2 · 2019-10-23
**Official Blind Review #2**

**Rating:** 8

**Review:**

The paper presents SEED RL, which is a scalable reinforcement learning agent. The approach restructure the interface / division of functionality between the actors (environments) and the learner as compared to the distributed approach in IMPALA (a state-of-the-art distributed RL framework). Most importantly, the model is only in the learner in SEED while it is distributed in IMPALA.

The architectural change from to IMPALA to SEED feels reasonable, and the results support the choices in a positive way.

SEED is evaluated using a large number of benchmarks using three environments, and the performance is compared to IMPALA. The results are very good, shows good scalability, and significantly reduced training times.

The paper is well written, easy to read, and I enjoyed it.

The code for SEED is released open source, which enables future research to build upon SEED.


**Experience Assessment:**

I have read many papers in this area.

**Review Assessment: Checking Correctness Of Derivations And Theory:**

I assessed the sensibility of the derivations and theory.

**Review Assessment: Checking Correctness Of Experiments:**

I assessed the sensibility of the experiments.

**Review Assessment: Thoroughness In Paper Reading:**

I read the paper at least twice and used my best judgement in assessing the paper.

---

> ### Author Response · Authors · 2019-11-11
> **Response**
>
> We thank the reviewer for the time, comments on the paper and the appreciation of open sourcing the content of the paper.

---

### Official Review · AnonReviewer3 · 2019-11-01
**Official Blind Review #3**

**Rating:** 6

**Review:**

The paper proposes a new reinforcement learning agent architecture which is significantly faster and way less costly than previously distributed architectures. To this end, the paper proposes a new architecture that utilizes modern accelerators more efficiently.  The paper reads very well and the experimental results indeed demonstrate improvement. Nevertheless, even though working in deep learning for years and have also some experience with Reinforcement learning I am not in the position to provide an expert judgment on the novelty of the work. I do not know if ICLR is the right place of the paper (I would probably suggest a system architectures conference for better assessment of the work).

**Experience Assessment:**

I do not know much about this area.

**Review Assessment: Checking Correctness Of Derivations And Theory:**

I assessed the sensibility of the derivations and theory.

**Review Assessment: Checking Correctness Of Experiments:**

I assessed the sensibility of the experiments.

**Review Assessment: Thoroughness In Paper Reading:**

I read the paper at least twice and used my best judgement in assessing the paper.

---

> ### Author Response · Authors · 2019-11-11
> **Support for including SEED at the ICLR conference**
>
> We thank the reviewer for the time and positive comments on the paper.
>
> To support including the paper at the ICLR conference, we note that ICLR in previous years included papers with similar flavor to SEED such as,
> Distributed Prioritized Experience Replay (Ape-X), ICLR 2018
> Recurrent Experience Replay in Distributed Reinforcement Learning (R2D2), ICLR 2019

---

### Official Review · AnonReviewer5 · 2019-11-01
**Official Blind Review #5**

**Rating:** 8

**Review:**

This paper presents a scalable reinforcement learning training architecture which combines a number of modern engineering advances to address the inefficiencies of prior methods. The proposed architecture shows good performance on a wide variety of benchmarks from ALE to DeepMind Lab and Google Research Football. Important to the community, authors also open source their code and provide an estimate which shows that the proposed framework is cheaper to run on cloud platforms.

Pros:
1. This work is solid from the engineering perspective. It effectively addresses the problems with prior architectures and the accompanying source code is clear and well structured. It is also extensively tested on several RL benchmarks.

2. The proposed framework is especially suited for training large models as the model parameters are not transferred between actors and learners.

3. The paper is well written and organized.

Cons:

1. The gain of the main algorithmic improvement (SEED architecture) over the baseline (IMPALA architecture) is obscured by the usage of different hardware. TPUv3 has different characteristics than Nvidia P100/V100 GPU chips which also might contribute to the speed up.

Questions:

1. Is it possible to provide more “apple-to-apple” comparison by running SEED and IMPALA on the same hardware (TPUv3 or Nvidia P100/V100 GPU)?

**Experience Assessment:**

I have read many papers in this area.

**Review Assessment: Checking Correctness Of Derivations And Theory:**

N/A

**Review Assessment: Checking Correctness Of Experiments:**

I assessed the sensibility of the experiments.

**Review Assessment: Thoroughness In Paper Reading:**

I read the paper thoroughly.

---

> ### Author Response · Authors · 2019-11-11
> **Regarding apple-to-apple comparison**
>
> We thank the reviewer for the time and the positive comments.
>
> With regards to comparing apples-to-apples, we will add the performance of running SEED with Nvidia P100’s. Note, the cost of running IMPALA does not improve significantly with TPUs as the cost is dominated by inference on CPU.

---

> > ### Author Response · Authors · 2019-11-15
> > **Paper updated with apple-to-apple comparison**
> >
> > We thank again the reviewer for their positive comments.
> >
> > We have updated the paper with an "apple-to-apple" comparison by running both agents on an Nvidia P100 GPU. See table 1 for update figures, as well as additional analysis in section 4.1.2 and additional cost comparison in section A.6.

---

### Decision · Program_Chairs · 2019-12-19

**Decision:**

Accept (Talk)

**Comment:**

The paper presents a framework for scalable Deep-RL on really large-scale architecture, which addresses several problems on multi-machine training of such systems with many actors and learners running.  Large-scale experiments and impovements over IMPALA are presented, leading to new SOTA results. The reviewers are very positive over this work, and I think this is an important contribution to the overall learning / RL community.